# Retrograde Approach to Maxillary Nerve Block: An Alternative in Orofacial Surgeries in Horses

**DOI:** 10.3390/ani12111369

**Published:** 2022-05-27

**Authors:** Matteo Vuerich, Sara Nannarone

**Affiliations:** 1Department of Veterinary Medicine, University of Perugia, 06126 Perugia, Italy; matteo.vuerich@hotmail.it; 2Sport Horse Research Centre (C.R.C.S.), University of Perugia, 06126 Perugia, Italy; 3Research Centre on Animal Pain (Ce.Ri.D.A.), University of Perugia, 06126 Perugia, Italy

**Keywords:** equine, infraorbital canal, infraorbital foramen, infraorbital nerve, maxillary nerve, regional anesthesia, Tuohy needle

## Abstract

**Simple Summary:**

Maxillary nerve (MN) desensitization can improve quality and safety of orofacial surgeries in horses, especially when a standing procedure is elected. The purpose of this retrospective study is to report results from 15 horses undergoing orofacial surgery which received an MN block with the retrograde approach within the infraorbital canal. The same technique was used in horses scheduled for orofacial surgeries and grouped as requiring general anesthesia or standing analgo-sedation. From the retrieved anesthetic records, regardless of the group, clinical and physiological parameters continuously monitored during surgery were stable, and none of the horses showed complications during or after the block. These results confirm the feasibility of this loco-regional anesthetic technique as a valid approach to the MN without risk of damaging periorbital structures, which is reported with traditional techniques. Evident landmarks for this approach and limited chances to injure vital structures enclosed within the infraorbital canal make the operator experience less relevant than the ability required to block the MN according to approaches, which currently aim at the pterygopalatine fossa. This substantiates the retrograde approach to block the MN is safe and easy to perform in horses undergoing orofacial surgeries at regions where this nerve provides sensory innervation.

**Abstract:**

The aim of this study was to retrospectively report outcomes resulting from the approach to the maxillary nerve block (MNB) through the infraorbital canal, in terms of needles selection, drawbacks or side effects during or after block execution, and analgesic efficacy leading to clinical and cardiovascular stability during surgery. Anesthetic records of 15 horses undergoing orofacial surgery in standing analgo-sedation (STA, *n* = 6) and in general anesthesia (GEN, *n* = 9) were retrieved and analyzed. Horses in group STA required surgery for dental extraction, nasal polyp resection and maxillary/frontal sinusitis. Horses in group GEN underwent surgery for fronto-maxillary, nasal and dental diseases. Size 19 and 20 G Tuohy needles were used in adult horses weighing 350–600 kg, while size 21 and 22 G were used in younger horses or ponies. None of the horses in both groups showed complications related to the block and physiological parameters were stable and within normal ranges during surgery; overall, an adequate anesthetic/sedation depth was achieved. Our results confirm the in vivo applicability of the MNB approached within the infraorbital canal, which had been described only on cadaveric specimens. The retrograde technique resulted in a valid and easy approach to the maxillary nerve that avoids damage to periorbital structures and side effects reported with traditional techniques.

## 1. Introduction

Perineural and locoregional nerve blocks are used in equine medicine for diagnostic and surgical purposes. Locoregional anesthesia should help to reduce pain perception, therefore improving horse welfare and compliance, the likelihood of a successful procedure as well as the safety for the operators, especially in standing animals. When performed in recumbent anesthetized horses, locoregional anesthesia contributes to pre-emptive analgesia, and therefore, to reduced requirements for inhalant anesthetic or systemic analgesics [1]. 

Appropriate regional analgesia of the equine head allows to increase safety during surgical standing procedures and to achieve more reliable and stable general anesthesia [1]. The maxillary branch of trigeminal nerve block is performed for surgical procedures that involve the ipsilateral upper teeth, maxillary bone, soft tissues, paranasal sinuses and nasal cavities and to investigate head shaker horses in trigeminal neuritis [2,3]. Commonly, several approaches allow blocking or desensitizing the maxillary nerve (MN) at the pterygopalatine fossa, where the MN enters through the maxillary foramen into the infraorbital canal (IC). Several different techniques have been described, including the caudolateral [3], the lateral [4] and the supraorbital [5] approaches. In particular, an equivalent accuracy has been demonstrated between the caudolateral and lateral approach, suggesting that the post mortem deposition of dye relative to the nerve could be complete, partial or missing [6]. According to several authors, desensitizing the MN is difficult, and therefore, strongly dependent on the operator experience, because the landmarks for injection are vague [6,7,8]

Moreover, the proximity of important vascular structures in this region [9] (deep facial vein, maxillary and descending palatine arteries) makes these approaches potentially dangerous. The risk of puncturing these vessels might lead to hematoma formation, temporary blindness and severe retrobulbar swelling [6,10]. The risk of introducing infection into retrobulbar tissues also exists [11]. The supraorbital approach involves the risk to inadvertently paralyze the extraocular muscles [5]. For all these reasons, ultrasonography seems advantageous over the blind technique [12]. Ultrasound guidance reduces complications rate, improves the quality of the block and might decrease the incidence of neuro- or cardiotoxicity consequent to unintentional intravascular injection or delayed tissue uptake [13].

Another alternative and safer solution to block the MN might be the retrograde administration of local anesthetic within the IC to the maxillary foramen. Wilkins et al. [14], in an attempt to block the infraorbital nerve for head-shaking diagnosis, suggested that this approach could be annoying and not well-tolerated by horses. From this statement, Nannarone et al., in 2016 [15], described a feasible approach to the maxillary nerve block (MNB) within the IC via the infraorbital foramen (IF), evaluating the anatomy of the canal and providing a correct needle selection in cadaveric skulls. However, one of the main limitations of the study was to have validated the technique on cadaver specimens.

The purpose of our study is to retrospectively report outcomes of the approach to the MNB through the IC in equine patients undergoing head surgery, and to add information to the previously described method [15], detailing benefits and precautions when the technique is applied in clinical cases. We hypothesized that the block would improve the sedation/general anesthetic depth and overall outcome, reducing isoflurane requirements, as well as the need for intra-operative systemic analgesics/anesthetics top-up and contributing to maintain vital parameters within physiological ranges.

## 2. Materials and Methods

This is a retrospective study consisting of analysis of anesthetic records of client owned horses referred to the Veterinary Teaching Hospital of Perugia University and scheduled for elective orofacial surgery between March 2014 and January 2021. The owners signed an informed consent for surgery and an agreement for the use of scientific data from their animals. Anesthetic records were manually retrieved from the archive. 

Horses were included in the study if they had received the retrograde MNB as loco-regional anesthesia and if the anesthetic record was complete in all its parts, including relevant time points, description of size and type of needle and any drawbacks during or after the procedure. Demographic data of the horses were also recorded. Horses were excluded if an MNB was not performed.

Among 20 retrieved anesthetic records of horses undergoing orofacial surgery, only 15 animals satisfied all inclusion criteria. There were 10 females, 4 geldings and 1 stallion; the age range was 38 days–22 years, and the weight range was 130–600 kg. Animals were grouped as undergoing standing surgery (group STA, *n* = 6) or general anesthesia (group GEN, *n* = 9), depending on the preference of the surgeon according to the type of the disease (Table 1). 

Each horse received a complete pre-anesthetic clinical evaluation, different antibiotic (dihydrostreptomycin, penicillin, gentamycin or ceftiofur) and anti-inflammatory (phenylbutazone or flunixin meglumine) were administered pre-operatively according to the case. 

The MNB was always performed by the same anesthetist and two equally experienced surgeons were involved in all surgeries.

All horses in the STA group were restrained in a stock with the head sustained by means of a headrest and sedated intravenously with detomidine (8.4 µg/kg) and methadone (0.1 mg/kg), followed by detomidine infused according to a previously described regimen [16]. A further 15–20 mL of 2% lidocaine could be provided by the surgeon as local infiltration with hypodermic needle at the site of skin incision, as well as a splash with 2% lidocaine + 0.002% noradrenaline (Lidocaina 2%, A.T.I. S.r.l., Ozzano dell’Emilia (BO), Italy) at the intended area. 

Horses in group GEN were premedicated intravenously with an association of alpha-2 agonist/opioid (romifidine 30–50 µg/kg or detomidine 10–12 µg/kg with methadone or morphine 0.1 mg/kg), then anesthesia was induced with diazepam/ketamine (0.04/2.2 mg/kg) and maintained with isoflurane and lidocaine infusion (30–50 µg/kg/min). 

All horses in both groups were instrumented with a multiparameter monitor (Beneview T5, Mindray, Shenzhen, China) for continuous recording of heart rate (HR) and arterial blood pressure (noninvasive or invasive technique in group STA and GEN, respectively), which were reported in the anesthetic record every 5 mins. The cuff for noninvasive blood pressure in horses belonging to group STA was placed at the coccygeal artery, but no correction according to heart distance was reported. The arterial line for invasive blood pressure in horses in group GEN was placed in a facial (*n* = 7) or in a metatarsal artery (*n* = 2). 

Beside physiological values, clinical parameters (i.e., palpebral reflex, nystagmus, limb movements or purposeful gross movement/kick in group STA or GEN, respectively) enabled to define a score for anesthesia/sedation depth, ranging from 0 to 2 (0 = deep, 1 = adequate, 2 = too light). 

Heart rate and mean arterial blood pressure (MAP) values retrieved from every horse have been collected and reported in an Excel file, after identification of specific time points: the basal time (considered as 5 mins before the MNB), the time of MNB, the time of start of surgery and every 5 mins thereafter. Mean value ± standard deviation for all time-points was calculated for the two groups. The recorded parameters were evaluated to verify any modification from basal values throughout surgery time (about 80 mins) to assess the stability of the anesthesia. To retrospectively evaluate the anesthetic depth, we referred to a scoring system often used for research purposes at our institution [17] and here adapted considering only MAP values. If a MAP 20% greater than the value recorded at basal time was noted, accompanied or not by a purposeful reaction (depth score = 2), anesthetic depth was considered inadequate speculating for inefficacy of the block.

Despite different anesthetic or sedation protocols, the technique for the MNB was the same in both groups and was performed as reported by Nannarone et al. [15], except that in all horses in group STA, a subcutaneous injection of 1–2 mL of 2% lidocaine was performed at the site of Tuohy needle introduction (see Table 1 for size and length).

In brief, the technique for the MNB consisted in locating the IF with the ‘three fingers grip’ technique, where the middle finger is placed on the arch of the naso-incisive notch, the thumb on the rostral end of the facial crest and the index finger elevates the ventral edge of the *levator labii superioris muscle*, palpating the IF (Figure 1). 

While displacing dorsally the infraorbital nerve with the same index finger, a Tuohy needle was gently inserted through the skin 2–3 mm rostral to the IF and into its ventral part (Figure 2, Figure 3 and Figure 4). Regardless of the local anesthetic, the injected volume was 10 mL in adult horses and 5 mL in foals for each MNB (see Table 1).

## 3. Results

Among the 20 protocols retrieved, 5 horses (*n* = 2 in general anesthesia and *n* = 3 in standing sedation) were excluded because no MNB was performed, for evidence of a compromised infraorbital canal at computed tomography (*n* = 1) or unreported reasons (*n* = 4). 

The retrograde technique within the IC was used to block 18 maxillary nerves in 15 horses scheduled for orofacial surgery. One horse received a bilateral surgery in two different occasions: at 38 days and at 3 years in group GEN and STA, respectively (Figure 3 and Figure 4).

The Tuohy needle was chosen essentially according to the horse size so that in those weighing less than 300 kg the smaller 21 and 22 G were preferred, while Tuohy needles sized 19 and 20 G were successfully used in horse between 350–600 kg. The only reported drawbacks were experienced with four horses as five Tuohy needles could only partially be inserted within the canal. This occurred in one horse (one 21 G × 8 cm needle) in group GEN and three horses (two 19 G × 8 cm and two 20 G × 9 cm needles, due to a bilateral surgery in a same animal) in group STA (Table 1). However, the introduced length of each partially inserted needle was greater than or equal to 5 cm. 

The type of injected local anesthetic differed among animals as reported in Table 1, while the volume was 10 mL in all adult horses and 5 mL in the two foals of group GEN for each MNB. 

None of the horses in groups STA and GEN had reported complications related to the block during or after its execution (i.e., hematoma formation, temporary blindness, retrobulbar swelling) and physiological parameters were stable and within normal ranges throughout surgery both during standing procedures (Figure 5) and general anesthesia (Figure 6). 

In group STA mean detomidine infusion was 0.2 ± 0.06 µg/kg/min. Three horses in this group had scores of 2 for sedation depth and the total dose of IV detomidine as top-up ranged from 1.7 to 5 µg/kg. The same three horses, two requiring a frontonasal flap and one having a dental repulsion, received a subcutaneous infiltration of 15–20 mL of 2% lidocaine at the site of skin incision and a further local splash of about 30 mL of a mixture of 2% lidocaine + 0.002% noradrenaline. 

Isoflurane requirement during surgery in horses in group GEN was 0.9–1.3%. 

The mean time elapsed from the completion of the block and the beginning of surgery differed between the two groups, resulting in a shorter time (15 ± 7 mins) when horses were under general anesthesia than in standing analgo-sedation (30 ± 16 mins) (Figure 7).

## 4. Discussion

Performing orofacial surgeries with the horse standing results in less hemorrhage than when the same surgeries are performed under general anesthesia [7]. Moreover, standing surgery eliminates the risk of general anesthesia, lowers costs and often is preferred by surgeons for their personal skills or technical execution, as the approach to the surgical field might be more direct and comfortable. For these reasons, providing a reliable regional anesthesia in the standing sedated horse should significantly contribute to a multimodal analgesic strategy to increase safety for horses, reducing single drug dosages and leading to more reliable and comfortable procedures for both animal and operators. 

This study reports the outcomes of the approach to block the MN within the infraorbital canal used in horses undergoing standing sedation or general anesthesia for orofacial surgeries. The success of the described technique was clinically assessed by evaluating horse tolerance to the procedure, in terms of lack of prompt reactions, especially during standing surgery, and adrenergic response during surgical manipulations regardless of the approach (i.e., standing or recumbent animals). According to data retrieved from the anesthetic records of only 15 horses, it is likely that the retrograde technique was easily performed and neither damage to periorbital structures nor perioperative side effects were reported. Several complications have been described when performing the MNB from periorbital approaches toward the pterygopalatine fossa. Lesions to the transverse facial and maxillary vessels may cause periocular or retrobulbar hematoma, collapse and blindness [6,10]. Periorbital injections could induce ocular globe prolapse [18], neurological and respiratory deficits or cardiac arrest if local anesthetic is injected into the dural cuff of the optic nerve, as suggested by Staszyk et al. [9], or could induce ataxia up to collapse if injected intravascularly [19]. Finally, the supraorbital approach involves the risk to inadvertently paralyze the extraocular muscles [5].

Ultrasound guidance could be very useful over the blind technique to reduce the complication rates [12]. According to several authors, the operator experience is essential to succeed in MNB using classical approaches that rely on proper orientation of the needle within specific and more or less evident anatomic landmarks [6,7,14].

The retrograde approach via the IF to desensitize the MN is safe and simple to perform. Only two small vascular structures and the infraorbital nerve are present within the IC. At the emergence of the infraorbital nerve the foramen is elliptical. This dorso-ventral long axis cross-section provides a safe and free ventral pathway for clinicians to introduce the Tuohy needle (which appeared superior to the Quincke needle given its anti-coring curve), reducing risks of vascular injuries or involuntary nerve puncture [15].

According to what was described on cadaver specimens [15], in brief, the ‘three fingers grip technique’ enables to identify the IF and to detect the infraorbital nerve by trans-cutaneous palpation. The IF can be further easily located at about 12 cm from a vertical line starting at the medial cantus of the ipsilateral eye toward the exploring index finger [15]. Sliding the index finger dorsally along the nerve and the *levator labii superioris muscle* can improve the gap between the ventral wall of the IF and the ventral side of the nerve, making this technique even safer. 

Evident landmarks and low risk to damage the structures included within the IC make the operator experience less important than the ability and knowledge required to block the MN according to supraorbital approaches.

As already reported by Wilkins [14], when blocking the infraorbital nerve to diagnose headshaking, the procedure might be not well tolerated by head-shy horses. In this study, to limit the possible stimulation of the infraorbital nerve and the consequent prompt head jerking during needle insertion, the horses in group STA were sedated, sometimes nasally twitched and a subcutaneous injection of 1–2 mL of 2% lidocaine at the site of needle introduction was performed. 

When needle insertion becomes difficult for an evident bony resistance, the authors recommend to (1) mildly retract the needle of approximately 0.5–1 cm, (2) to rotate the needle orienting the bevel slightly ventrally (clockwise in the left IC and counter-clockwise in the right IC) and (3) to reinsert it. If obstruction persists in standing horses, a forced advancement should be avoided to limit any trigger to violent head reactions and consequent possible canal fractures. Nevertheless, it is important that the Tuohy needle is inserted within the canal for at least 5 cm so that a volume of 10 mL local anesthetic may reach or pass the maxillary foramen [15]. Anatomical modifications due to bone defects or due to the underlying pathological conditions, which might have altered the normal path of the canal, or the physiological course of the IC, which presents a deviation after about 4–5 cm from the initial linear path [15], may have prevented needle progression into the canal at first attempt. Moreover, differences exist between young and adult horses due to maturation of the dental arcade and maxillary sinus, and this should be taken into account when selecting needle size [15,20]. For this reason, small-sized 22 G × 5 cm needles, which are more flexible, have been preferred in two foals in group GEN, without facing any drawbacks. In this retrospective study, only five Tuohy needles (8 and 9 cm long) were partially inserted: one in group GEN and four in group STA. The horse undergoing standing nasal septum reconstruction and bilateral rhinoplasty was the same experiencing frontal bone fracture repair when it was a foal. This pathologic condition might have implied a possible anatomic modification of the ICs justifying the incomplete progression of the needles at both sides during the standing procedure. However, the inserted length of each needle was always greater than or equal to 5 cm, confirming this as the minimum insertion depth within the IC still adequate to produce an effective nerve block. This was demonstrated by stable cardiovascular parameters (i.e., HR and MAP) and lack of score 2 for sedation/anesthetic depth during surgical manipulation, regardless of the group, except on three horses in group STA. Any increase of HR and or MAP, accompanied or not by a purposeful reaction (score 2), would suggest an adrenergic response due to painful perception. It is reported that for surgical procedures involving the paranasal sinuses and the nasal cavity, a maxillary nerve block will in some instances not be sufficient, due to sensory innervation of these structures by the ophthalmic branch of the trigeminal nerve and an additional block of the ethmoidal nerve can be beneficial in these cases [10,21]. This is likely the reason why in those horses in group STA requiring a frontonasal flap, a subcutaneous infiltration of 2% lidocaine was used to grant desensitization of a skin area otherwise innervated essentially by the frontal nerve. 

In group GEN, low isoflurane requirements may suggest that a consistent contribution to anesthetic depth was provided by lidocaine infusion and perineural anesthesia. 

Nevertheless, a limitation of the study is the low number of animals and that no real control group, either without MNB or lidocaine infusion, is available for comparison. However, it is likely that the perineural block would contribute to decrease intraoperative nociception and reduce central sensitization. 

With this retrospective study, we aimed to report the technical feasibility and clinical usefulness of the approach within the infraorbital canal to perform an MNB in horses scheduled for orofacial surgery. For this reason, the influence of different anesthetic or analgesic drugs used both systemically and locally was beyond our scope. However, three different local anesthetic agents have been used either alone or combined, and this was mainly due to the anesthetist’s choice. Regardless of the molecule, the volume of injected local anesthetic for each MNB was 10 and 5 mL in adults and foals, respectively. 

When selecting a local anesthetic, the aim should be to find the right compromise between onset and duration of action according to the type of the planned surgery. Onset of action is faster for lidocaine and mepivacaine (5 to 10 mins) compared to bupivacaine (15–45 mins). Given the higher vasodilator effect of lidocaine, it results in a shorter effect (60–120 mins) compared to both mepivacaine (90–180 mins) and bupivacaine (180–480 mins), with the latter being highly lipid-soluble, and therefore, slowly washed out from the nerve membranes [22]. It is evident that the tendency to mix two local anesthetics (one intermediate and one long-acting agent) in all horses in the STA group was related to the belief of a desirable shorter onset time and longer effect. However, neural blockade performed mixing two local anesthetics is unpredictable and controversial, and it depends on the types of drugs and the pH of the mixture. Mixing lidocaine and bupivacaine may produce a faster and longer neural block [23]; conversely, no clinical advantage, with respect to onset and duration of local blockade, resulted when using a 50/50 mixture of lidocaine and bupivacaine in place of unmixed drugs in humans [24]. Moreover, in standing horses of our study, the time elapsed from the block to skin incision was about 30 mins (which would be a sufficient time for plain bupivacaine to work), which is about double the time elapsed in horses under general anesthesia. It is likely that this longer phase could be mainly due to the different time required to prepare the surgical field in a standing or recumbent animal. Surgical field preparation includes clipping and scrubbing of the area with antiseptics solution. In recumbent anesthetized horses, this is normally initiated as soon as the animal is properly positioned on the surgical table, while the anesthetist sets up monitoring and stabilizes the animal. On the other hand, in standing horse, the MNB is normally performed as soon as the horse appears sedated and after simple skin disinfection of the area at the intended IF, with or without clipping; then, the surgical field is aseptically prepared after the block, as it would be in recumbent horses. In the authors’ opinion, 15–20 mins after the block would be an adequate time before initiating skin incision in standing horses. 

## 5. Conclusions

According to the retrieved anesthetic records of 15 horses receiving a retrograde MNB as part of their analgesic plan, no complications or subsequent side effects were described during or after the nerve block. The retrograde MNB, unlike the supraorbital approaches, does not require a great knowledge of the anatomy and specific skills of the operator. Furthermore, the risks of damaging important structures with serious consequences are minimal.

When the technique is performed on standing animals, we recommend adequate sedation, proper physical restraint including nasal twitch if necessary and a subcutaneous injection of 1–2 mL of 2% lidocaine at the site of Tuohy needle introduction. It is important to isolate the infraorbital nerve by moving it dorsally along with the *levator labii superioris muscle* with the index finger and sliding in the Tuohy ventrally for at least 5 cm of its length, with the bevel lateral, which lowers the chance of triggering an animal reaction by touching the nerve.

These simple precautions should make this retrograde approach to the MNB a valid support for standing surgeries, such as the extraction of maxillary teeth or sinus diseases, even in a field condition. This would avoid hospitalization into referral clinics, thereby considerably reducing costs, while making the procedures safer and more feasible for many surgeons.

## Figures and Tables

**Figure 1 animals-12-01369-f001:**
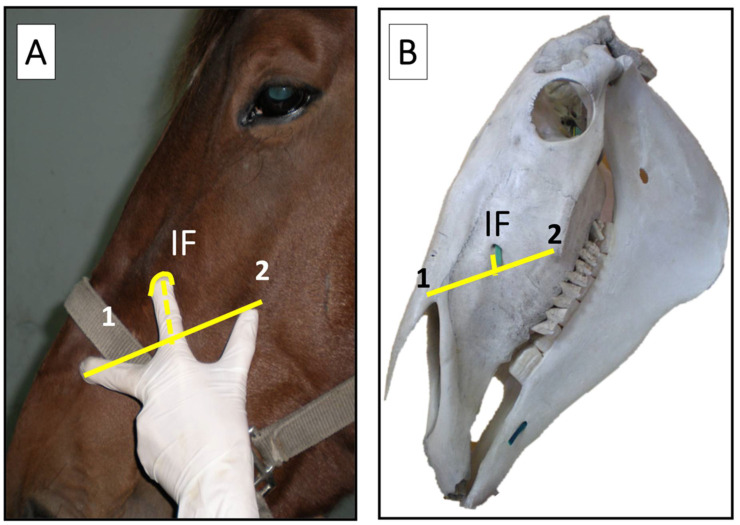
The ‘three fingers grip’ technique in vivo (**A**) and anatomical landmarks in a skull (**B**) used to localize the infraorbital foramen (IF), which lies below the *levator labii superioris muscle*. 1: naso-incisive notch; 2: facial crest.

**Figure 2 animals-12-01369-f002:**
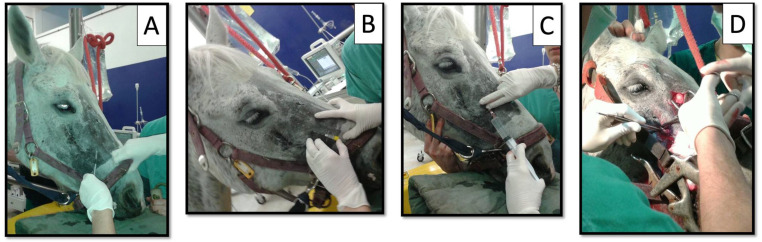
March 2014: a 17-year-old mare in group STA undergoing maxillary sinus trephination and dental root curettage. Horse was restrained in a stock, sedated and with the head sustained by a headrest. Subcutaneous injection of 1–2 mL 2% lidocaine at the infraorbital foramen (**A**); sliding of a 20 G × 5 cm Tuohy needle into the infraorbital canal with the bevel oriented laterally (**B**); injection of 10 mL local anesthetic after withdrawn of the stylet and needle aspiration to exclude any vessel puncture (**C**); surgical procedure (**D**).

**Figure 3 animals-12-01369-f003:**
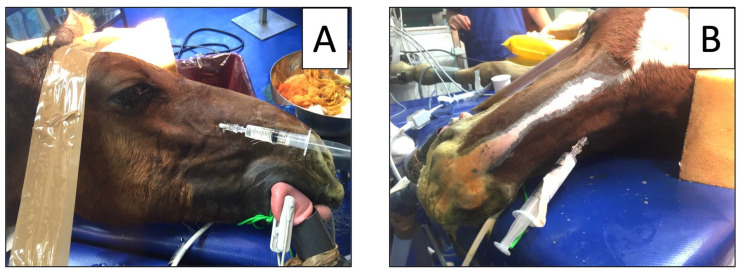
July 2016: a 38-day old foal in group GEN undergoing bilateral rhinoplasty for nasal septum fracture. Two 22 G × 5 cm Tuohy needles were inserted at the right (**A**) and left (**B**) infraorbital canal for bilateral block of the maxillary nerve.

**Figure 4 animals-12-01369-f004:**
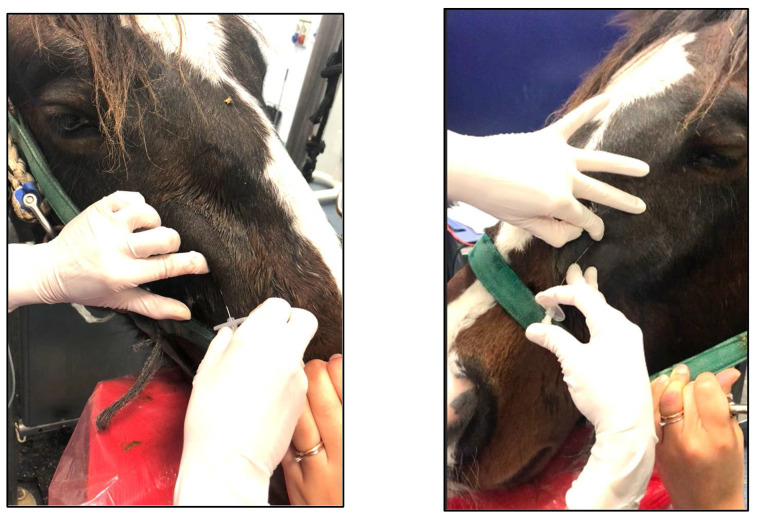
December 2019: a 3-year old horse in group STA undergoing bilateral rhinoplasty and nasal septum reconstruction. Particular of the insertion of a 20 G × 9 cm Tuohy needle into each infraorbital canal while elevating the infraorbital nerve and the *levator labii superioris muscle* with the index finger.

**Figure 5 animals-12-01369-f005:**
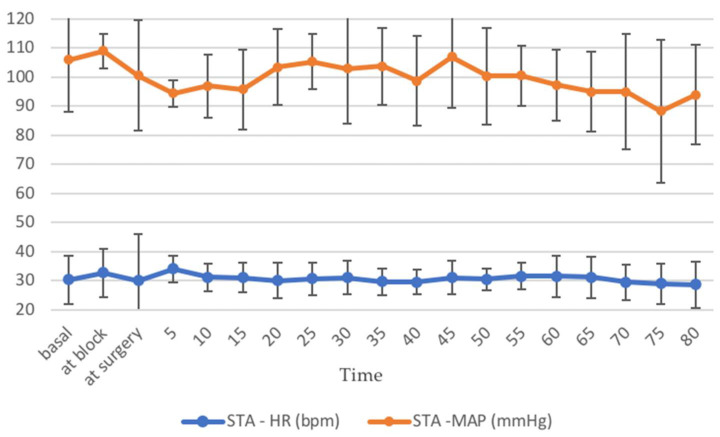
Heart rate (HR, beats/min) and mean arterial blood pressure (MAP, mmHg) recorded in horses included in group STA (undergoing standing analgo-sedation, *n* = 6) during the procedure. Data are reported as mean value and bars represent the standard deviation.

**Figure 6 animals-12-01369-f006:**
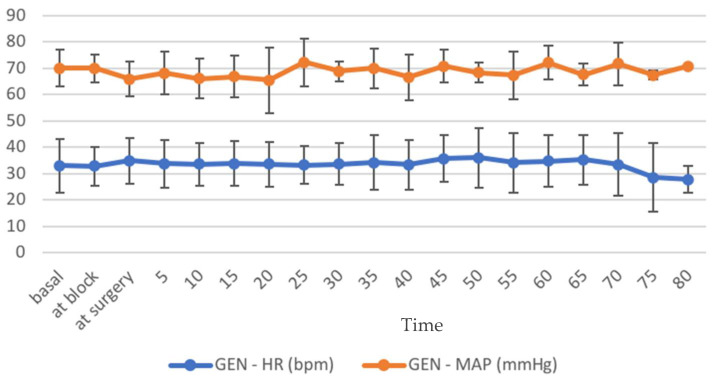
Heart rate (HR, beats/min) and mean arterial blood pressure (MAP, mmHg) recorded in horses included in group GEN (undergoing general anesthesia, *n* = 9) during surgery. Data are reported as mean value and bars represent standard deviation.

**Figure 7 animals-12-01369-f007:**
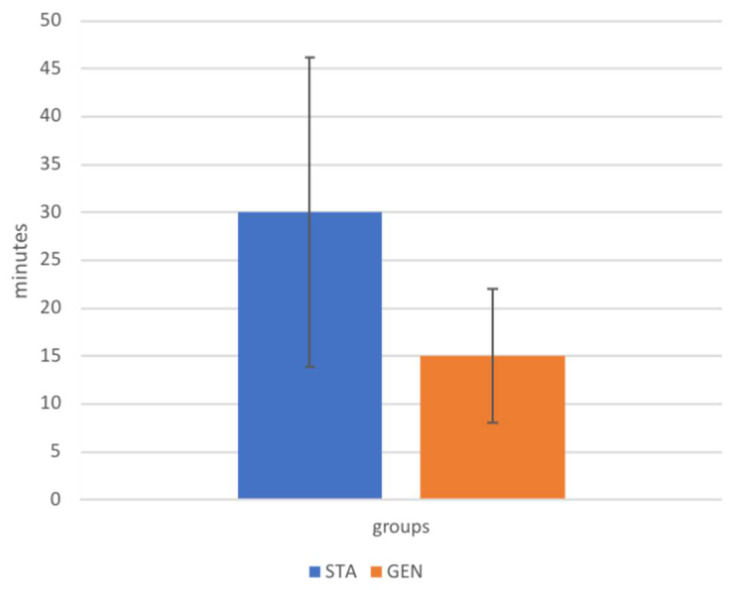
Time elapsed from block execution to start of surgery in horses included in group STA (undergoing standing analgo-sedation, *n* = 6) and group GEN (undergoing general anesthesia, *n* = 9). Data are reported as mean ± standard deviation.

**Table 1 animals-12-01369-t001:** Description of horses undergoing orofacial surgery which received a retrograde maxillary nerve block. STA = standing surgery; GEN = general anesthesia; * age is in years unless otherwise reported, i.e., in days (d) or months (m); B = 0.5% bupivacaine; L = 2% lidocaine; M = 2% mepivacaine; WB = Warmblood; BWP = Belgian Warmblood; KWPN = Royal Warmblood Studbook of the Netherlands; STB = Standardbred.

Group	Breed	Weight (kg)	Age (*)	Surgery	Tuohy Size Gauge × cm	Local Anesthetic	Drawbacks	Anesthesia Duration (mins)	Surgery Duration (mins)
STA	Italian WB	480	17	Rostral and caudal maxillary sinuses trephination, dental root curettage	20 G × 5	B + L	-	151	91
	Hannover	600	8	Frontonasal flap + nasal polyp resection	19 G × 8	B	Only 7 cm inserted	203	156
	KWPN	598	14	Frontonasal flap	19 G × 8	B +L	Only 6 cm inserted	90	75
	Italian WB	465	6	Maxillary flap + dental repulsion	21 G × 9	B + M	-	140	90
	BWP	470	3	Nasal septum reconstruction, bilateral rhinoplasty	20 G × 9	B + L	Left side 7 cm, right side 5 cm inserted	218	136
	French Mountain	540	4	Dental extraction	20 G × 5	B + L	-	145	115
	BWP	550	14	Frontonasal sinusotomy	20 G × 5	B	-	160	100
	Pony	290	7	Dental extraction	21 G × 8	B	-	120	105
	BWP	550	15	Frontonasal flap	21 G × 8	B	Only 7 cm inserted	190	143
	BWP	570	22	Frontonasal flap + dental	18 G × 8	B	-	110	70
	STB	350	1.5	Nasal membrane resection	19 G × 8	B	-	80	30
GEN	BWP	130	38 d	Nasal septum fracture, bilateral rhinoplasty	22 G × 5	B	-	133	70
	Italian WB	560	13	Conchofrontal sinus cyst, frontonasal flap	20 G × 9	B	-	134	90
	Purebred Arabian breed	174	6 m	Nasal ductus bilateral	22 G × 5	B	-	77	65
	Welsh Pony	220	17	Maxillary flap + dental extraction	22 G × 5	B	-	100	65

## Data Availability

The data presented in this study are available on request from the corresponding author.

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
