# Peer review of "Retrograde Approach to Maxillary Nerve Block: An Alternative in Orofacial Surgeries in Horses"

_animals, 2022, doi:10.3390/ani12111369_

Round 1
Reviewer 1 Report
Good paper.
This is a retrospective study. Records of cases were retrieved and analyzed for horses requiring oro-facial surgery. Fifteen animals were included in this study. Standing (n=6) and general anaesthetic (n=9).
Line 84-101: Describes the sedation/analgesia/general anaesthesia protocol. I understand that these are clinical cases and were carried out by different teams. It might be beneficial to provide a range (minimum-maximum) dose of the drugs used for the protocols. This so that people who routinely sedate/anaesthetise horses can understand the level of sedation required and dose of drugs used for general anaesthesia and the rate of CRI for lidocaine.
Line 94: Mentions an extra 15-20 ml of lidocaine provided (was it in some horses?) at the site of the skin incision. Why was this necessary if the local block was successful?
Line 97: Mentions lidocaine CRI in horses under general anaesthesia. Why was this necessary if you had carried out a local block?
I am curious as to how did they test in standing horses whether the block had worked and it was time to start surgery?
Line 255-284: Discusses the local anaesthetics, onset and duration. Table 1: Shows that bupivacaine was used in horses requiring general anaesthesia. Bupivacaine has a longer onset and duration. Surgery in horses under general anaesthesia commenced just under 15 minutes after the block. No changes in heart rate or mean arterial blood pressure were observed? How do you explain this?
Perhaps you might need to consult an veterinary anaesthetist who can go over the anaesthetic charts and assist you in determining the success of the local blocks.
Figures: I found the images in the pdf file I downloaded to be fuzzy and not sharp! I am assuming that you will have high quality images in the publication?
Line 10-11: ‘The purpose of the present study was to describe a new safe approach to the maxillary nerve block and to report its successful application in 15 horses undergoing orofacial surgery’.
From the data provided I find it difficult to draw this conclusion (see earlier comments). Its just not that you were able to insert a needle into the infraorbital foramen and inject a local anaesthetic, but whether that local anaesthetic was sufficiently able to block the maxillary nerve and provide a pain free area for surgery.
Author Response
Good paper.
This is a retrospective study. Records of cases were retrieved and analyzed for horses requiring oro-facial surgery. Fifteen animals were included in this study. Standing (n=6) and general anaesthetic (n=9).
Line 84-101: Describes the sedation/analgesia/general anaesthesia protocol. I understand that these are clinical cases and were carried out by different teams. It might be beneficial to provide a range (minimum-maximum) dose of the drugs used for the protocols. This so that people who routinely sedate/anaesthetise horses can understand the level of sedation required and dose of drugs used for general anaesthesia and the rate of CRI for lidocaine.
Dear reviewer, drugs and dosages have been included as requested at lines 111-117, moreover, according also to rev. 2, a description of monitoring and rules for the retrospective assessment has been reported.
Line 94: Mentions an extra 15-20 ml of lidocaine provided (was it in some horses?) at the site of the skin incision. Why was this necessary if the local block was successful?
Dear reviewer, thank you for pointing this out as we realize that we forgot an important description due to peculiarity of innervation of some anatomic structures of the head. We included more details about those horses in the standing group which received a subcutaneous infiltration of lidocaine at lines354-356. Anesthetizing the maxillary nerve at the level of the pterygopalatine fossa will desensitize the ipsilateral dental structures of the maxilla and premaxilla, the paranasal sinuses, and the nasal cavity. However, it has been reported that this nerve block does not always seem to completely desensitize the ipsilateral paranasal sinuses and nasal cavity from failure to completely anesthetize the maxillary nerve, or from stimulation of sinonasal areas innervated by the ethmoidal nerve. The ethmoidal nerve provides sensory innervation to the rostral portion of the frontal sinus, dorsal nasal concha, portions of the nasal mucosa, and the caudal portion of the nasal septum. This is why, as we included in this revision at the end of the discussions, the subcutaneous infiltration was especially provided in those horses where a frontonasal flap was elected by the surgeons (however a supraorbital/frontal nerve block might have benefit as well to desensitize the skin area where this flap is normally made, differently from the maxillary flaps). Finally, the desired technique used by our surgeons is that described by Schumacher and Perkins (Surgery of the Paranasal Sinuses Performed with the Horse Standing. Clin Tech Equine Pract 2005, 4:188-194) which also describe the use of local infiltration at the surgical site.
Line 97: Mentions lidocaine CRI in horses under general anaesthesia. Why was this necessary if you had carried out a local block?
Given the fact that this is a retrospective evaluation and not a clinical prospective study, we did not aim at assessing the beneficial effect of the block in reducing isoflurane requirements. As routine in our VTH, we aim to optimize perioperative analgesia and we always use multimodal pain management, so the block provides pre-emptive analgesia (together with the preanaesthetic drugs) then we further aim to deliver continuous analgesia matching the analgesic plan to the degree of surgery, this is provided by lidocaine in this case (or other anaesthetics/analgesic infusions). Volatile agents induce dose-dependent cardiovascular and respiratory depression in horses. Given the fact that the decrease in the amount of volatile agents required by patients undergoing general anesthesia is beneficial, we constantly aim to produced isoflurane-sparing effect, that’s why we always use PIVA protocols in our animals and any other form of multimodal approach for analgesia, as a perineural block in these cases. Nevertheless, we try to specify this concept at lines 296-301 and 343-345.
I am curious as to how did they test in standing horses whether the block had worked and it was time to start surgery?
Indeed, no real test was applied, if you intend for instance a pinprick, but surgeons relied only upon both asking the anaesthetist, if it was the right time for starting, and the horse reaction to the approach at the surgical area. Likewise, we must remember that, as it happens for human dentistry, a patient may remain disturbed by a sense of pressure despite complete anesthesia of pain fibers. It is likely that some horse, although sedated, may slightly react by mildly moving the head, at simple first looking at the surgeon’s hands approaching with drapes around the face or with the scalpel at the skull close to the eye. However, it does not necessarily mean that there’s still nociception as this should be coupled by a brisk reaction of the horse and an adrenergic response at the time of skin incision.
Line 255-284: Discusses the local anaesthetics, onset and duration. Table 1: Shows that bupivacaine was used in horses requiring general anaesthesia. Bupivacaine has a longer onset and duration. Surgery in horses under general anaesthesia commenced just under 15 minutes after the block. No changes in heart rate or mean arterial blood pressure were observed? How do you explain this?
Dear reviewer, this section is now from line 309. Except for 4 recumbent horses in which surgery started around 10’ from the block, in the other 5 the elapsed time was indeed superior (the figure has been amended after more deeply looking at the symbols for timing included in the records, apologies for the first mistake) ranging anyhow from 10-25’ – but general anaesthesia was started since a mean time of 32±15 minutes; we could speculate that the multimodal analgesic protocol somehow have contributed to a surgical anaesthetic depth or simply that the cardiovascular values recorded did not exactly coincided with skin incision as these could be written within 5 minutes time (according to the normal way of recording anaesthetic protocols every 5 minutes) but it is not easy to extrapolate now from the retrieved records.
Perhaps you might need to consult an veterinary anaesthetist who can go over the anaesthetic charts and assist you in determining the success of the local blocks.
Dear reviewer, the anesthetic records have been graphically summarized as mean±SD of the values of the essential cardiovascular parameters (mean blood pressure and heart rate) which were monitored during general anaesthesia (and standing) and reported every 5’. Values from each horse were reported in an excel file for final evaluation and then to obtain the graphs. This is to say that we went through the records (which always also report any eventual movement which was considered accordingly!) and attained to what is traditionally considered, that is, cardiovascular responses are used as indicators of nociception during anaesthesia. An unstable anaesthetic depth is normally referred as an adrenergic response followed or not by sudden movement/reaction of animals.
We included the rationale for this retrospective assessment, according to literature and rules commonly followed at our VTH especially when performing research studies. We considered any incremental of MAP greater than 20% from values recorded at the time of block as a possible index of inadequate analgesia and ineffective nerve block (121-122, 184-188, and 294-301) and further implemented the discussion with this concept at lines 337-349.
Figures: I found the images in the pdf file I downloaded to be fuzzy and not sharp! I am assuming that you will have high quality images in the publication?
Figures have been changed with those at improved quality, thank you for noticing it.
Line 10-11: ‘The purpose of the present study was to describe a new safe approach to the maxillary nerve block and to report its successful application in 15 horses undergoing orofacial surgery’.
From the data provided I find it difficult to draw this conclusion (see earlier comments). Its just not that you were able to insert a needle into the infraorbital foramen and inject a local anaesthetic, but whether that local anaesthetic was sufficiently able to block the maxillary nerve and provide a pain free area for surgery.
Dear review, we agree with your comments and according also to reviewer 2, as reported in earlier answers, we included a better description on how we evaluated the anaesthetic records to judge the efficacy of the block intraoperatively (121-122, 184-188, and 294-301) and further implemented the discussion with this concept at lines337-349.

Reviewer 2 Report
Dear authors
The manuscript is interesting and written fairly well. My main criticism is that it is mentioned in the discussion and conclusion that the technique investigated is effective, but the methods are not clearly designed to assess efficacy, and the results do not support this statement in this form. Of course I do not question the efficacy of the technique, as it would not have been possible to perform the standing surgery! I believe you need to slightly amend the manuscript to justify clearly why you consider the technique effective, especially in anaesthetised horses. You somehow introduce the topic suggesting that the monitored parameters demonstrate the efficacy of the block, but you do not say how and it is up to the reader to work it out.
The other important piece of information is the length of the needle. Tuohy needles of the same diameter come in different length, so you should specify the length of the needles used.
Now some specific comments:
- line 17: probably “without risk of damaging periorbital structure, which is reported…” is better than “that avoids the risk of damage to …”
- Lines 18-21: this sentence is quite long and not very clear, in particular the concept of experience being “less mandatory” - do you mean less relevant?
- Line 24: “may” is better than “confer risks to”
- Line 25 “results of” can be removed as not necessary
- Line 33: “Size 19 and 20 G Touchy needles were used in adult horses weighting 350-600 kg, sizes 21 and 22 G in younger ones or ponies.”
- Line 45: “An” can be deleted
- Line 47: “planes of” can be deleted
- Line 49: “the” before “head shakers” can be deleted
- Line 77: “The purpose of the study is…”
- Line 78: replace “surgery for pathologies at the head” with “head surgery”
- Line 82: change “had benefit of the” to “may benefit from”
- Line 96” “then anaesthesia was induced”
- Lines 97-99: I would change to “Beside the anaesthetic or sedative protocol, the technique for the nerve block was the same in both groups…”
- Line 114-119: this is where it is crucial to say how long were the needles you used to perform the block
- In the methods session you should also specify the monitoring used- ie how did you measure blood pressure in the standing horse? Did you use an ECG as well? Was the monitoring under sedation and anaesthesia equivalent?
- Line 204: please consider finding an alternative to “mandatory”
- Line 205: change “through” to “using”’.“relay” is incorrect spelling- should be “rely”
- Line 219” “poor chance to injure” is a bit awkward- probably “low risk to damage” is better
- Line 220: change “mandatory” - maybe “important”?
- Line 225: your retrospective study is not designed in a way to directly demonstrate efficacy. In the discussion you should try to bring forward a strong argument demonstrating the efficacy of the technique. While this is relatively easy for the standing surgery, it may be more tricky for the procedures performed under anaesthesia, given you will probably have to speculate on anaesthetic requirements without having a control group.
- Line 273: “plain bupivacaine” is the correct spelling
- References:
- Ref 10 does not follow the number correctly (ie it appears for the first time after ref 15
- Some references may be removed (for example, you have lots of generic references on blocks of the head)
- Fig 7: the caption does not say how data are reported (I assume mean and SD)
Author Response
Dear authors
The manuscript is interesting and written fairly well. My main criticism is that it is mentioned in the discussion and conclusion that the technique investigated is effective, but the methods are not clearly designed to assess efficacy, and the results do not support this statement in this form. Of course I do not question the efficacy of the technique, as it would not have been possible to perform the standing surgery! I believe you need to slightly amend the manuscript to justify clearly why you consider the technique effective, especially in anaesthetised horses. You somehow introduce the topic suggesting that the monitored parameters demonstrate the efficacy of the block, but you do not say how and it is up to the reader to work it out.
Dear reviewer, thank you for your general appreciation, we agree that the effectiveness of the block was poorly addressed, and a better description is now included already from M&M at lines 117-122, then in the results at lines 184-188, and in the discussion 294-301 and further implemented with this concept at lines 346-350.
The other important piece of information is the length of the needle. Tuohy needles of the same diameter come in different length, so you should specify the length of the needles used.
This was actually reported in the table, now we simply include a mention to specifically see the table at the point you’ve suggested (line 145). This is due to the fact that unfortunately several sizes and length were used during those surgeries probably according to the available needles at that time.
Now some specific comments:
- line 17: probably “without risk of damaging periorbital structure, which is reported…” is better than “that avoids the risk of damage to …”
Thank you for your suggestion, the text has been amended.
- Lines 18-21: this sentence is quite long and not very clear, in particular the concept of experience being “less mandatory” - do you mean less relevant?
Correct, text has been modified at line 21; that is to say that with this technique you are not required to know exactly the due direction to follow when inserting a needle with a periorbital approach from caudal, ventral and dorsal to the bony orbit, orientating the needle dorsoventrally, lateromedially or rostromedially in order to avoid important structures.
- Line 24: “may” is better than “confer risks to”
Amended, thank you
- Line 25 “results of” can be removed as not necessary
Text has been removed, according to your suggestion, thank you
- Line 33: “Size 19 and 20 G Touchy needles were used in adult horses weighting 350-600 kg, sizes 21 and 22 G in younger ones or ponies.”
Amended at line 35-36, thank you
- Line 45: “An” can be deleted
Amended, thank you
- Line 47: “planes of” can be deleted
Amended, thank you
- Line 49: “the” before “head shakers” can be deleted
Amended, thank you
- Line 77: “The purpose of the study is…”
Amended at line 88, thank you
- Line 78: replace “surgery for pathologies at the head” with “head surgery”
The sentence has been corrected at line 87, thank you
- Line 82: change “had benefit of the” to “may benefit from”
Amended at line 100, thank you
- Line 96” “then anaesthesia was induced”
Amended at line 115, thank you, all the sentence has been implemented according to requests from reviewer 1.
- Lines 97-99: I would change to “Beside the anaesthetic or sedative protocol, the technique for the nerve block was the same in both groups…”
The sentence has been changed according to your suggestion at line 122-123, thank you
- Line 114-119: this is where it is crucial to say how long were the needles you used to perform the block
Now this point is at line 145 and a reference to table 1 for size and length has been included in parenthesis. In fact, given the different needles adopted (probably simply due to length available at that time) the only relevant point would probably be a suggestion to enter at least for 5 cm but this was reported in the discussion already, and we included a sentence in the conclusions at lines 336-337.
- In the methods session you should also specify the monitoring used- ie how did you measure blood pressure in the standing horse? Did you use an ECG as well? Was the monitoring under sedation and anaesthesia equivalent?
Thank you for pointing this out, your suggestion to include description of monitoring has been included in the M&M section at lines 117-122. We used the same multiparameter monitor in all horses, but in standing horses only noninvasive blood pressure (not the invasive one) was measured together with HR.
- Line 204: please consider finding an alternative to “mandatory”
The term mandatory has been changed into essential at line 246, thank you.
- Line 205: change “through” to “using”’.“relay” is incorrect spelling- should be “rely”
The words have been amended at line 247, thank you
- Line 219” “poor chance to injure” is a bit awkward- probably “low risk to damage” is better
Amended at line 262, thank you
- Line 220: change “mandatory” - maybe “important”?
Amended at line 262, thank you.
- Line 225: your retrospective study is not designed in a way to directly demonstrate efficacy. In the discussion you should try to bring forward a strong argument demonstrating the efficacy of the technique. While this is relatively easy for the standing surgery, it may be more tricky for the procedures performed under anaesthesia, given you will probably have to speculate on anaesthetic requirements without having a control group.
Dear reviewer, thank you for pointing out this important lack of description on how we evaluated the anaesthetic protocols to assess adequacy of the anaesthetic depth and consequent efficacy of the block, also considering that we did not have any control group (either without nerve block or without lidocaine CRI). We included the rationale for this retrospective assessment, according to literature and other studies undertaken by our group. We considered any incremental of MAP greater than 20% from values recorded at the time of block as a possible index of inadequate analgesia and ineffective nerve block (121-122, 184-188, and 294-301) and further implemented the discussion with this concept at lines 346-350.
Line 273: “plain bupivacaine” is the correct spelling
Amended at line 327, thank you
- References:
- Ref 10 does not follow the number correctly (ie it appears for the first time after ref 15
We apologize for this mistake. The order has been corrected according also to your following comment.
- Some references may be removed (for example, you have lots of generic references on blocks of the head)
According to this comment, we removed 4 references (n.1 Oel et al. 2014; n.17 Schumacher et al. 1994; n.18 Quinn et al. 2005; n.19 Barakzai and Dixon 2014), we modified Skarda and Tranquilli 2007 into Labelle and Clark-Price 2013, and we included two new refs.: Nannarone et al. 2011 and Caruso et al. 2016.
- Fig 7: the caption does not say how data are reported (I assume mean and SD)
Thank you for noticing this mistake, the caption has been amended
Round 2
Reviewer 2 Report
Dear authors
thanks for addressing my questions.
I have just some minor editing suggestions related to style/typos/English:
- line 10 and 12: remove "the" before "maxillary
- line 26: remove "," prior to "may"
- lines 32 and 34: to be honest, probably it is not relevant to add weight and age in the abstract, so I would remove them to make it read better
- line 63: remove "The risk of"
- line 86: remove "to" before "standing"
- line 25: "progress was impeded" does not read well. "resistance was encountered" sounds better
- Table 1: this is important, apologies for missing in the previous review! Please explain the abbreviations used for the breeds
- line 156: remove "in each group", as it is the same horse that underwent standing and GA procedure in two different occasions, not two different horses
- line 162-166: this is methods, not results, should be added after line 106 (after the sentence you added in the revised manuscript, ending in "throughout the surgery")
- line 187: "greater than or equal to"
- line 255: change "could be the cause of prevented" to "may have prevented"
- line 256: "Moreover,"
- line 259: change "favored" to "preferred"
- line 260: "series, only"
- line 262: "greater than or equal to 5 cm, "
- line 265: change "coupled" to "accompanied" and "eventual" to "a"
- line 290: change "elicited by" to "performed"
- line 294: change "their independent use" to "unmixed drugs"
- line 296: change "available time for the plain bupivacaine" to "sufficient time for plain bupivacaine to work"
- line 306: "the block, "; add "15-20 minutes"; maybe enquiry with the editor if "mins" is an accepted abbreviations, as in therms of word count it does not make any different using "minutes" or mins", and using "minutes" may be easier
- line 308: "perception, "
- line 309: "operators, "
- line 310: "animals, "
- line 311: "analgesia, "
- line 313: remove "achieving local anaesthesia via", as it is redundant
- line 318: "sufficient, "
- line 329: "performed in standing animals"; change "well sedated patients" to "adequate sedation"
- line 335: remove "itself"
